# Genetically Engineered Hepatitis C Virus-like Particles (HCV-LPs) Tagged with SP94 Peptide to Acquire Selectivity to Liver Cancer Cells via Grp78

Dina Mofed [1], Mohamed A. Wahba [1] and Tamer Z. Salem [1,2,3,*]

1 Molecular Biology and Virology Lab, Biomedical Sciences Program, UST, Zewail City of Science and Technology, October Gardens, 6th of October City, Giza 12578, Egypt
2 Department of Microbial Genetics, Agricultural Genetic Engineering Research Institute (AGERl), ARC, Giza 12619, Egypt
3 National Biotechnology Network of Expertise (NBNE), Academy of Science Research and Technology (ASRT), Cairo 11334, Egypt
* Correspondence: tsalem@zewailcity.edu.eg

**Abstract:** Targeted cancer therapy is a challenging area that includes multiple chemical and biological vehicles. Virus-like particles (VLPs) combine safety and efficacy in their roles as potential vaccines and drug delivery vehicles. In this study, we propose a novel drug delivery system based on HCV-LPs engineered with SP94 and RGD peptides mediated by a specific molecular chaperone (Grp78) associated with cancer drug resistance. The PCR primers were designed for engineering two constructs, SP94-EGFP-CORE-HIS and RGD-EGFP-CORE-HIS, by sequential PCR reactions. The two fragments were cloned into pFastBac Dual under the polyhedrin promoter and then used to produce two recombinant baculoviruses (AcSP94 and AcRGD). The VLP's expression was optimized by recombinant virus infection with different MOIs, ranging from 1 to 20 MOI. Recombinant VLP2 were purified by Ni-NTA and their sizes and shapes were confirmed with TEM. They were incubated with different types of cells prior to examination using the fluorescence microscope to test the binding specificity. The effect of the overexpression of the Grp78 on the binding affinity of the engineered VLPs was tested in HepG2 and HeLa cells. The protocol optimization revealed that MOI 10 produced the highest fluorescence intensities after 72 h for the two recombinant proteins (SP94-core and RGD-core). Moreover, the binding assay tested on different types of mammalian cells (HeLa, HEK-293T, and HepG2 cells) showed green fluorescence on the periphery of all tested cell lines when using the RGD-core protein; while, the SP94-core protein showed green fluorescence only with the liver cancer cells, HepG2 and HuH7. Overexpression of Grp78 in HepG2 and HeLa cells enhanced the binding efficiency of the engineered VLPs. We confirmed that the SP94 peptide can be specifically used to target liver cancer cells, while the RGD peptide is sufficiently functional for most types of cancer cells. The overexpression of the Grp78 improved the binding capacity of both SP94 and RGD peptides. It is worth noting that the SP94 peptide can function properly as a recombinant peptide, and not only as a chemically conjugated peptide, as heretofore commonly used.

**Keywords:** virus-like particle; HCV; cancer; SP94 peptide; RGD; Grp78; drug delivery

## 1. Introduction

Conventional cancer treatments such as curative resection, transarterial chemoembolization, radioembolization, radiofrequency ablation, and systemic targeted agents, can be destructive, risking severe side effects on all dividing cells [1]. There are various biomedical applications in drug delivery such as inorganic nanoparticles, liposomes, and polymers; however, no clear-cut solution is available and each has some limitation, such as biocompatibility, deliverability, and specificity [2]. Virus-like particles (VLPs) have emerged as

new drug vehicles to reduce the deleterious effect of many chemotherapies and also allow several conventional drugs to be administrated in less harmful ways [3].

VLPs are made by altering the viruses' genomes to become nucleic-acid-free particles. They can be loaded with anti-cancer drugs and conjugated with cancer-targeting peptides [4]. VLPs are metastable; they can endure environmental stress but are sensitive to cellular stimulations leading to cargo dislodgement. One of the most prominent mammalian-derived VLPs is the hepatitis B virus; it is heavily investigated both as a vaccine and as a gene vehicle. In addition, recent work showed that hepatitis E VLPs has intrinsic tissue tropism to the liver and can enter via endocytosis to deliver plasmids [5], however, little is known about hepatitis C VLPs.

In a previous study, the HCV core protein fused with RGD peptide (arginine-glycine-aspartic) and the IFN-$\alpha$2a protein, which was expressed by the baculovirus expression system, inhibited the migration and invasion of the breast cancer cells MDA-MB231 [6]. RGD peptides can bind to most integrins, which are highly expressed in cancer cells, especially the $\alpha$v$\beta$3 integrin, therefore RGD can be effective in delivering anti-tumor drugs to most types of cancer cells [6,7]. Integrins are heterodimeric transmembrane (TM) proteins, which belong to cell adhesion receptors and contain two non-covalently associated $\alpha$ and $\beta$ subunits. They control many biological functions such as growth, migration, and survival of cells; they also have an important role in the invasion and migration of tumor cells [8].

On the other hand, SP94 (SFSIIHTPILPL), a novel peptide identified by Lo et al. using a phage display technique, can specifically bind to various types of liver cancer cells, such as HepG2 and Huh-7 [9]. VLPs of bacteriophage MS2 displaying SP94 were loaded with different types of anti-cancer drugs such as doxorubicin, cisplatin, and 5-fluorouracil in order to kill HCC cell lines selectively [10].

The 78-kDa glucose-regulated protein (Grp78), also known as BiP/HSP5a, is a member of the heat shock proteins family (HSP70). Grp78 is known to be located in the lumen of the ER to help in protein folding, however, it is found in several other different places, such as cell membranes. Recent findings suggest that, cell surface Grp78 (CS-GRP78) is accumulated in stressed cells and cancer cells, and especially in drug-resistant cells. Grp78 is one of the main reasons for drug resistance and the recurrence of the tumor; it plays an important role in the invasion, proliferation, and metastasis of cancer [11]. It has been suggested that Grp78 is a receptor for SP94. The Ferritin Fn (HccFn) nanocage that displayed the SP94 peptide could be a promising delivery carrier for an anti-HCC drug without causing damage to healthy tissues [12]. In this study, HCV-LPs were engineered from core 173 and fused with EGFP and tumor targeting peptides RGD or SP94 to acquire a specificity toward cancer cells. In addition, we highlighted the role of Grp78 when overexpressed in HepG2 and HeLa cells, confirming that Grp78 is the receptor mediating the binding of the SP94 peptide.

## 2. Materials and Methods

### 2.1. Construction of VLPs Tagged with RGD and SP94 Cancer-Specific Peptides

Primers were designed for building two constructs namely SP94-EGFP-CORE-HIS and RGD-EGFP-CORE-HIS. Each construct was cloned in two steps, SP94-EGFP or RGD-EGFP, and then Core-His.

(A) SP94-EGFP was constructed after three sequential PCR reactions using Phusion High Fidelity DNA Taq polymerase (ThermoFisher Scientific, Waltham, MA, USA), using specific primers listed in (Table 1). The first PCR amplicon was amplified by primers 1 and 7; then it was purified by a gel purification kit (Qiagen, Hilden, Germany). The same process was followed with the sequential PCR reactions using primers 2 and 7 and finally by primers 3 and 7 to produce the final amplicon. The same procedures for RGD-EGFP were conducted, but by using primers 4 and 7, then primers 5 and 7, and finally primers 6 and 7. The purified PCR products of SP94-EGFP and RGD-EGFP were digested with restriction endonucleases *Bam*H1- and *EcoR1* (New England Biolabs-NEB, Ipswich, MA, USA); the same enzymes were used to cut

the plasmid pFastBac Dual (ThermoFisher Scientific, Waltham, MA, USA). Digested products were purified and ligated with T4 DNA ligase (New England Biolabs-NEB, Ipswich, MA, USA) to obtain pFast-SP94-EGFP and pFast-RGD-EGFP constructs. The ligated products were transformed into competent *E. coli* DH5a, as in our previous work [13].

(B)  For the second cloning fragment, HCV core-His was constructed by two sequential PCR reactions using Phusion High Fidelity DNA Taq polymerase (ThermoFisher Scientific, Waltham, MA, USA); (a) First PCR amplicon was amplified by primers 8 and 9, then primers 8 and 10. The purified amplicon was digested with restriction endonucleases *EcoR1* and *Xbal* (New England Biolabs-NEB, Ipswich, MA, USA); the same enzymes were also applied to plasmids pFast-SP94-EGFP and pFast-RGD-EGFP. Digested products were purified and then ligated with T4 DNA ligase (New England Biolabs-NEB, Ipswich, MA, USA) to form pFast-SP94-EGFP-core-His (pSP94-core) and pFast-RGD-EGFP-core-His (pRGD-core). The ligated product transformed into competent *E. coli* DH5a as indicated above.

**Table 1.** List of primers used in sequential PCR to build the two constructs.

| Item | Primer | Sequence of Primer |
|------|--------|--------------------|
| 1 | SP94_linker_F | 5′ CTTTAGCATTATTCATACCCCGATTCTGCCGCTGGGAGGTGGAGGA 3′ |
| 2 | SP94_linker_EGFP_F | 5′ CGCTGGGAGGTGGAGGAGTGAGCAAGGGCGAGGAG 3′ |
| 3 | BamH1_SP94_F | 5′ GCGGATCCGCCACCATGGTGAGCTTTAGCATTATTCATACCCCGA |
| 4 | RGD_linker_F | 5′ ACCATGGTG CGTGGCGATGGAGGTGGAGGA 3′ |
| 5 | RGD_linker_EGFP_F | 5′ GCGAT GGAGGTGGAGGAGTGAGCAAGGGCGAGGAG 3′ |
| 6 | BamH1_RGD_F | 5′ GCGGATCCGCCACCATGGTGCGTGGCGAT |
| 7 | EcoR1_EGFP_R | 5′ GCGAATTCCTTGTACAGCTCGTCCATG 3′ |
| 8 | EcoR1_Core_F_ | 5′ GCGAATTC AGCACGAATCCTAAACCT 3′ |
| 9 | Core_linker_R | 5′ ATGATGTCCTCCACCTCCTCCGGAGCAACCGGGGAGATT 3′ |
| 10 | linker_xba1_R | 5′ GCTCTAGATTAATGGTGATGGTG ATGATGTCCTCCACCTCCTCC 3′ |

## 2.2. Construction of Recombinant Baculoviruses

The two transfer vectors (pSP94-core and pRGD-core) were digested by restriction endonucleases *EcoR1* and *Xbal* (New England Biolabs-NEB, Ipswich, MA, USA) to confirm the presence of the genes before validation by nucleotide sequencing (Eurofins Genomics, Ebersberg, Germany). The pSP94-core and the pRGD-core were used to transform DH10Bac competent cells to produce bacmids, AcSP94, and AcRGD, all via Tn7-mediated transposition following the "bac-to-bac manual" (ThermoFisher Scientific, Waltham, MA, USA). These recombinant bacmids were verified by PCR before being transfected into Sf9 cells to produce the recombinant viruses.

## 2.3. Sf9 Transfection

EX-CELL 420 insect media (Sigma Aldrich, St. Louis, MO, USA), supplemented with 10% Fetal Bovine Serum (FBS) (Gibco, ThermoFisher Scientific, Waltham, MA, USA), was used to sustain the *Spodoptera frugiperda*-Sf9 insect cell line in a monolayer at 26 °C. For transfection, Sf9 cells were seeded in a 6-well-plate at a density of $8 \times 10^5$ cells per well before being incubated for 1 h at RT to attach. The transfection mixture was prepared by mixing 5 μg of DNA from each bacmid and 7 μL Cellfectin-II reagent (Invitrogen, ThermoFisher Scientific, Waltham, MA, USA) to a final volume of 200 μL serum-free media. The master mix and DNA were mixed thoroughly by vortex and added dropwise to the cells, after removing the media. The plate was left on the rocker for 4 h at RT. The transfection mixture was removed and replaced with 2 mL of fresh media before incubation at 26 °C. Cells were examined after 48–72 h post-transfection by Zoe fluorescent cell imager (BioRad, Hercules, CA, USA). The two recombinant viruses were amplified to P2 and the concentration was measured by end-point dilution assay as described in [14].

The protein expression was optimized by infection of Sf9 cells in a 24-well plate at density ($2 \times 10^5$ cell/well) at different Multiplicity of Infection (MOI) of 1, 5, 10, and 20 for 48, 72, and 96 hpi.

### 2.4. Purification and Identification of HCV-LP

Sf9 cells were seeded in a 6-well plate at a density of $8 \times 10^5$ cells per well and infected with two recombinant viruses (AcSP94 and AcRGD) at MOI of 10 for 48, 72, and 96 hpi then checked by Zoe fluorescent cell imager (BioRad, Hercules, CA, USA). The culture media was removed and cells were mixed with 200 µL of cell lysis buffer (CelLytic M, Sigma-Aldrich, St. Louis, MO, USA) and then incubated for 5 min on a rocking platform shaker. The cell's lysate was collected and centrifuged at ~14,000× $g$ for 10 min at 4 °C. The supernatant containing the VLPs was collected and the protein concentration was measured using a Pierce BCA Protein Assay kit (Thermo Fischer).

For in-gel fluorescence, 50 µL of the crude VLPs, collected at 48, 72, and 96 hpi, were mixed with 10 µL 4× of SDS-free sample buffer and separated on 10% sodium dodecyl sulfate-polyacrylamide gel (SDS-PAGE) by electrophoresis for 2 h at 100 volts according to [15].

The image was taken by ChemiDoc XRS+, with AlexFluor 488 filter (BioRad, Hercules, CA, USA). For protein purification, VLPs were collected at 72 hpi and purified by affinity chromatography using Ni-NTA agarose, according to the manufacturer's protocol (Qiagen, Hilden, Germany). Transmission electron microscope (TEM) negative staining was used to characterize the morphology of the purified VLPs by loading on a carbon-coated Cu-grid (200 mesh) for 5 min at RT and negatively stained with 2% phosphotungstic acid for 3 min before air drying on a filter paper. The samples were examined using a transmission electron microscope (Jem-2100, USA) at 200 KeV.

### 2.5. Binding Assay

HeLa and HepG2 cells were cultured in RPMI, while Huh7 and HEK-293T cells were cultured in DMEM high glucose (Gibco, ThermoFisher Scientific, Waltham, MA, USA), supplemented with 10% fetal bovine serum (FBS), and 100 U/mL penicillin G, and 100 µg/mL streptomycin, (Gibco, ThermoFisher Scientific, Waltham, MA, USA) at 37 °C and 5% $CO_2$. All cells were treated and incubated with 100 µM of either SP94-core or RGD-core for 2 h at 37 °C. Then, cells were washed with phosphate buffer saline (PBS) (Sigma-Aldrich, St. Louis, MO, USA) 2–3 times and examined with Leica DMi8 inverted microscope (Leica Microsystems, Wetzlar, Germany) at a magnification of 20×.

### 2.6. Overexpression of Grp78 in HeLa and HepG2 Cells

HepG2 and HeLa cells were cultured in RPMI (Gibco, ThermoFisher Scientific, Waltham, MA, USA) medium, supplemented with 10% FBS, 100 U/mL penicillin G, and 100 µg/mL streptomycin (Gibco, ThermoFisher Scientific, Waltham, MA, USA), before being incubated at 37 °C with 5% $CO_2$. Cells were seeded at a density of 100,000–150,000 cells per 30-mm cell culture petri dish for transfection with BiP-mCherryplasmid (Addgene; plasmid #62233) or pPAmCherry-a-tubulin plasmid (Addgene; plasmid#31930). Lipofectamine 3000 transfection reagent was used for transfection (ThermoFisher Scientific, Waltham, MA, USA), and 2.5 g DNA for each vector was added to the cells. Before adding the transfection complexes to cells, they were prepared in Opti-mem reduced serum medium (ThermoFisher Scientific, Waltham, MA, USA) and cultured at RT for 30 min. The transfected cells were incubated for 48 h at 37 °C with 5% $CO_2$. After 48 h of transfection, the transfected media were removed and cells were washed with PBS before being incubated with SP94-Core or RGD-Core proteins at a concentration of 100 µM for 2 h. Cells were washed with PBS 2–3 times and examined by Leica DMi8 inverted microscope (Leica Microsystems, Wetzlar, Germany) at a magnification of 100×.

### 3. Results

#### 3.1. Expression and Validation of SP94-Core and RGD-Core Proteins

SP94-EGFP-core-His and RGD-EGFP-core-His constructs were cloned in pFastbac dual to obtain the two transfer vectors: pSP94-core and pRGD-core (Figure 1A). The two recombinant vectors were confirmed on a 1% agarose gel after double digestions with *Bam*H1/*EcoR1*, *EcoR1*/*Xba1*, and *Bam*H1/*Xba1* to obtain the sizes of the expected fragments (Figure 1B). Both vectors, pSP94-core and pRGD-core, were authenticated by DNA sequencing before being transformed into DH10Bac to produce Bacmids via Tn7-mediated transposition, which eventually produced AcSP94 and AcRGD viruses after transfection into Sf9 insect cells (Figure 1A). The endpoint dilution was used to measure the titers of the produced recombinant virus, which were $9.4 \times 10^{11}$ and $4.48 \times 10^{11}$ pfu/mL for AcSP94-VLPs and AcRGD-VLPs, respectively. To select the most efficient day post-infection to harvest the expressed proteins, Sf9 cells were infected with MOI 10 and total proteins were isolated from cells at 48, 72, and 96 hpi. The in-gel fluorescence results confirmed that the highest fluorescence intensities were obtained at 72 hpi for both recombinant viruses (Figure 1C). In addition, TEM revealed spherical VLPs of 30 to 40 nm in size for RGD-VLPs and 40 to 60 nm for SP94-VLPs, which is the expected size range of the HCV virions (Figure 1D,E).

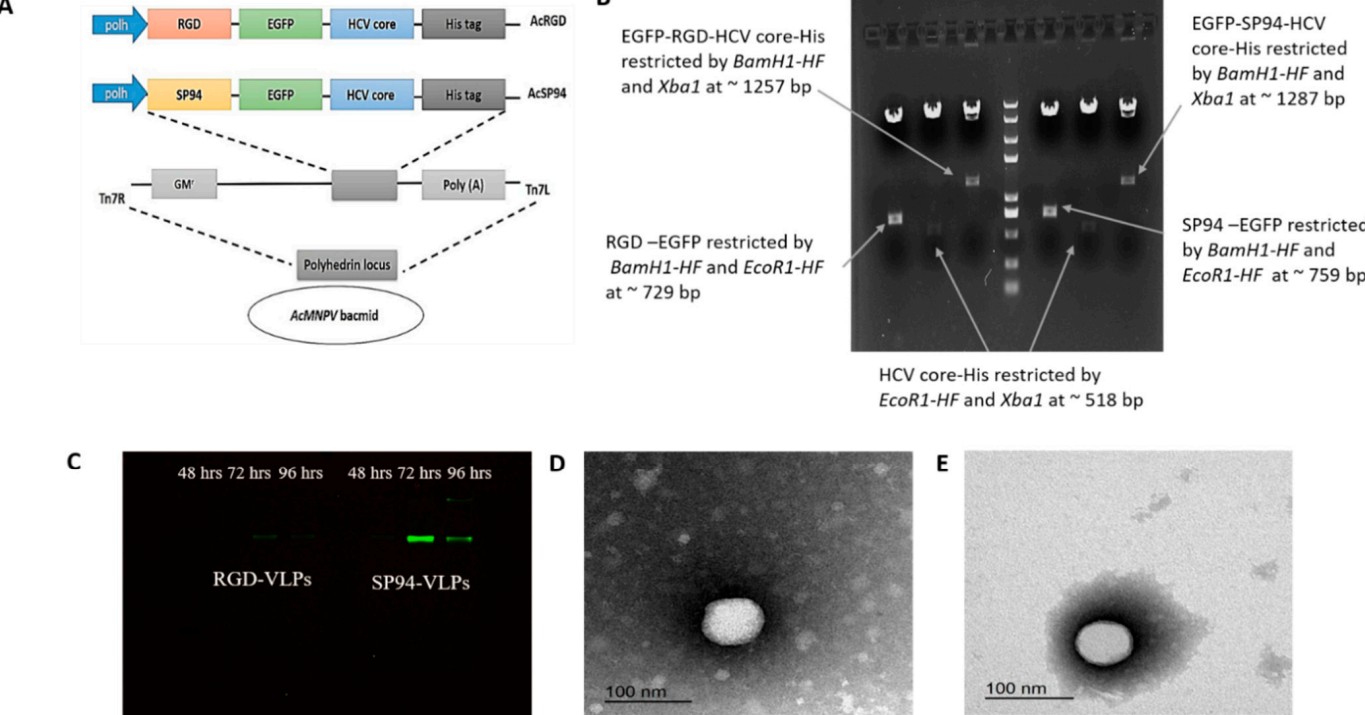

**Figure 1.** Construction and characterization of SP94-core protein and RGD-core protein: (**A**) represents the strategy for insertion of the two cloning cassettes into the polyhedrin locus of the AcMNPV bacmid; both cassettes were inserted into the attb site (indicated by the right and left insertion sites, Tn7R and Tn7L) in the polyhedrin locus by Tn-based transposition to generate the recombinant bacmids; (**B**) represents the digestion results of pSP94-core and pRGD-core by different restriction enzymes; (**C**) represents in-gel fluorescence results for SP94-core protein at 47.86 KDa and RGD-core protein at 47.86 KDa; (**D**) TEM of RGD-core protein; (**E**) TEM of SP94-core protein; TEM showed the VLPs sizes at 30 nm to 60 nm, which is the expected range of HCV.

#### 3.2. RGD-Core Protein Demonstrated Specificity to Cancer Cells

To prove the binding specificity of RGD peptide to cancer cells, RGD-core protein was incubated with different types of cells, including: HeLa, HepG2, Huh7, and HEK-293T cells for 2 h at a concentration of 100 µM. The results revealed that, as expected, the recombinant

RGD-core protein can specifically bind with all tested cancer cells. The specificity was demonstrated by the increase in green fluorescence intensity due to binding affinity after incubation with RGD-core. It was found that the RGD-core protein was highly bound to HeLa and Huh7 cells and slightly bound to HepG2 and HEK-293T cells (Figure 2).

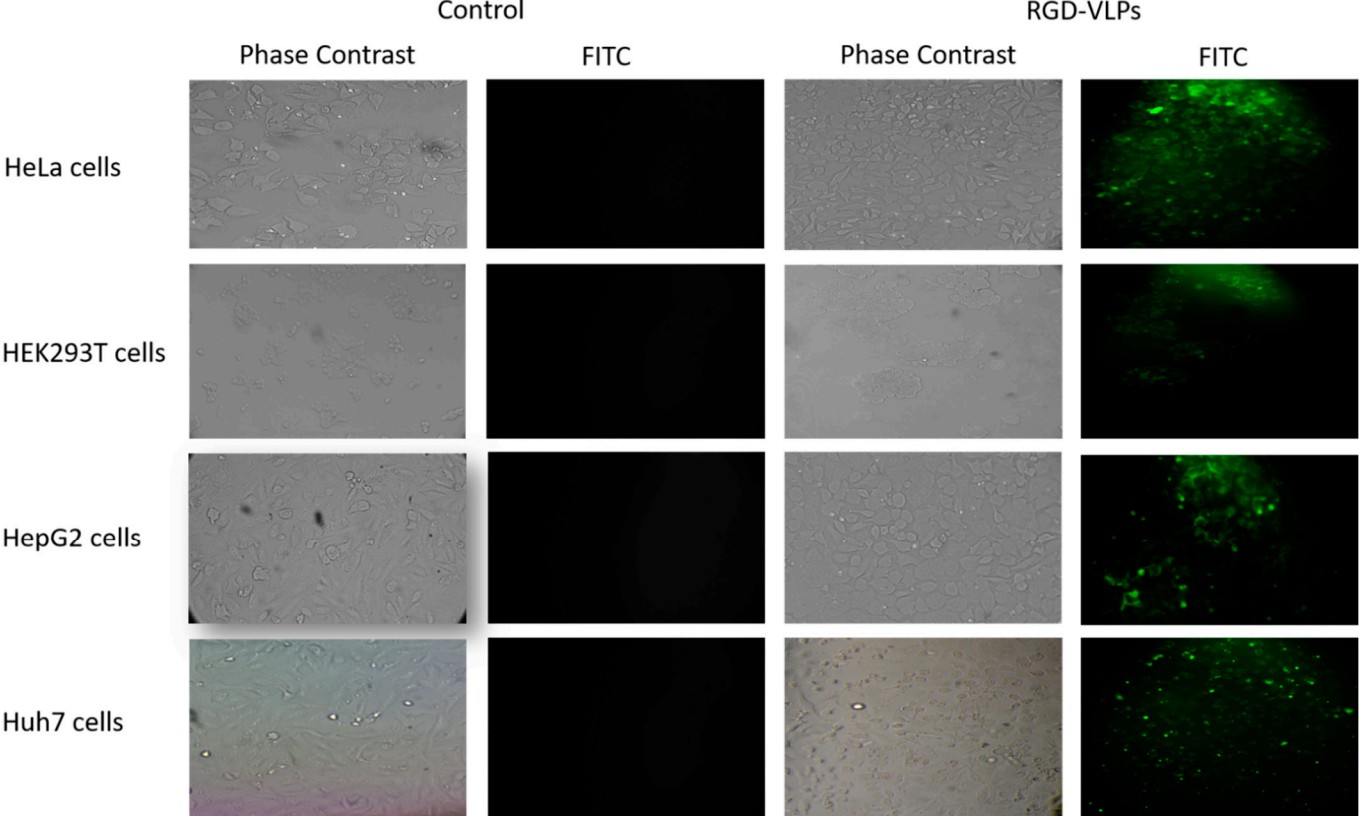

**Figure 2.** HeLa, HEK-293T, HepG2, and Huh7 cells treated with purified RGD-core protein fused with EGFP at a concentration of 100 μM for 2 h showed a high affinity to tumorigenic cells compared to none or low tumorigenic cells. Controls are cells not incubated with RGD-core.

### 3.3. SP94-Core Protein Demonstrated Specificity to Liver Cancer Cells

To confirm the specificity of SP94 peptide to liver cancer cells, SP94-core protein was incubated with different types of liver cells like HepG2 and Huh7 for 2 h at a concentration of 100 μM. Non-liver cells such as HEK-293T and HeLa cells were also used. The results showed that SP94-core was specifically bound to liver cancer cells HepG2 cells and Huh7 cells and not to non-liver cancer cells. It is worth noting that SP94-core protein exhibited an increase in green fluorescence intensity and a binding affinity with Huh7 cells than HepG2 (Figure 3).

### 3.4. Overexpression of Grp78 Increased Binding Efficiency of RGD-Core Protein to Cancer Cells

To assess the role of Grp78 as a possible marker for targeting cancer cells and its role in the enhancement of RGD-core protein binding, Grp78-mCherry and tubulin-mCherry were overexpressed in HeLa cells and HepG2 cells and incubated with RGD-core protein for 2 h at a concentration of 100 μM. The results demonstrated a high binding affinity of RGD-core protein, corresponding to the dramatic increase in the green fluorescence intensity in both types of cancer cells transfected with Grp78-mCherry compared to cells that were transfected with tubulin-mCherry. (Figure 4).

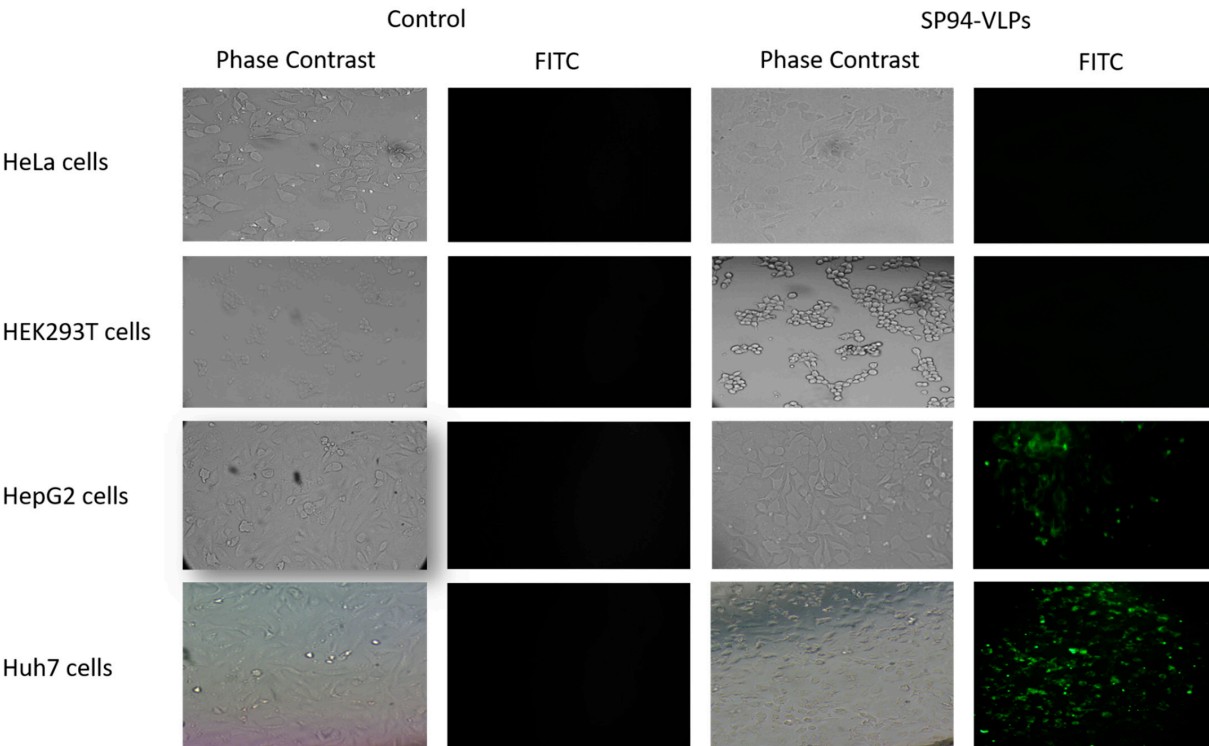

**Figure 3.** HeLa, HEK-293T, HepG2, and Huh7 cells treated with purified SP94-core protein at a concentration of 100 μM for 2 h, indicating the specificity only towards liver cancer (Huh7 and HepG2) compared to non-liver cells (HEK-293T and HeLa cells). Controls are cells not incubated with SP94-core.

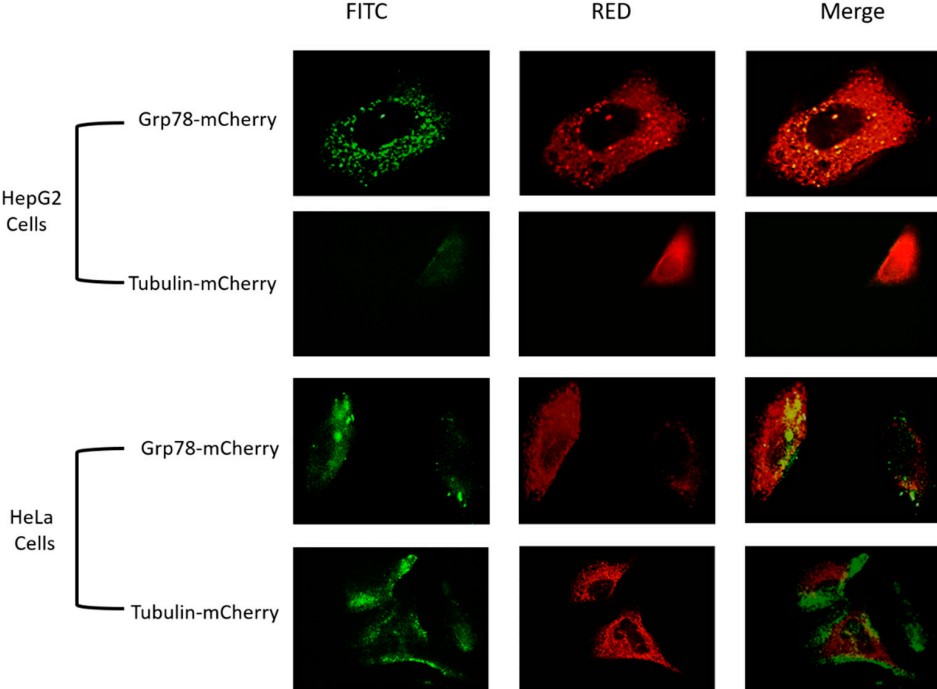

**Figure 4.** Enhancement of RGD-core binding affinity by Grp78 overexpression. HepG2 and HeLa cells were transfected with Grp78-mCherry and tubulin-mCherry (Red), then incubated with RGD-core protein at a concentration of 100 μM for 2 h. The results revealed an increase in the efficiency of binding affinity for cells transfected with Grp78-mCherry compared to cells transfected with tubulin-mCherry at the magnification of 100×.

### 3.5. Overexpression of Grp78 Increases Binding Efficiency of SP94-Core Protein

To prove the role of Grp78 in the enhancement of binding affinity of SP94-core protein, Grp78-mCherry and tubulin-mCherry were expressed in HepG2 cells and HeLa cells and incubated with SP94-core protein for 2 h at a concentration of 100 μM. The results revealed an obvious increase in green fluorescence intensity in HepG2 cells following overexpression of Grp78 compared to cells that transfected with a plasmid expressing tubulin-mCherry or to cells that incubated only with SP94-core protein (Figure 5). HeLa cells transfected with Grp78 and incubated with SP94-core protein showed green fluorescence intensity lower than that detected with HepG2 cells. However, no fluorescence was detected in HeLa cells that were only incubated with SP94 or transfected with tubulin-mCherry before being incubated with SP94-core protein (Figure 5).

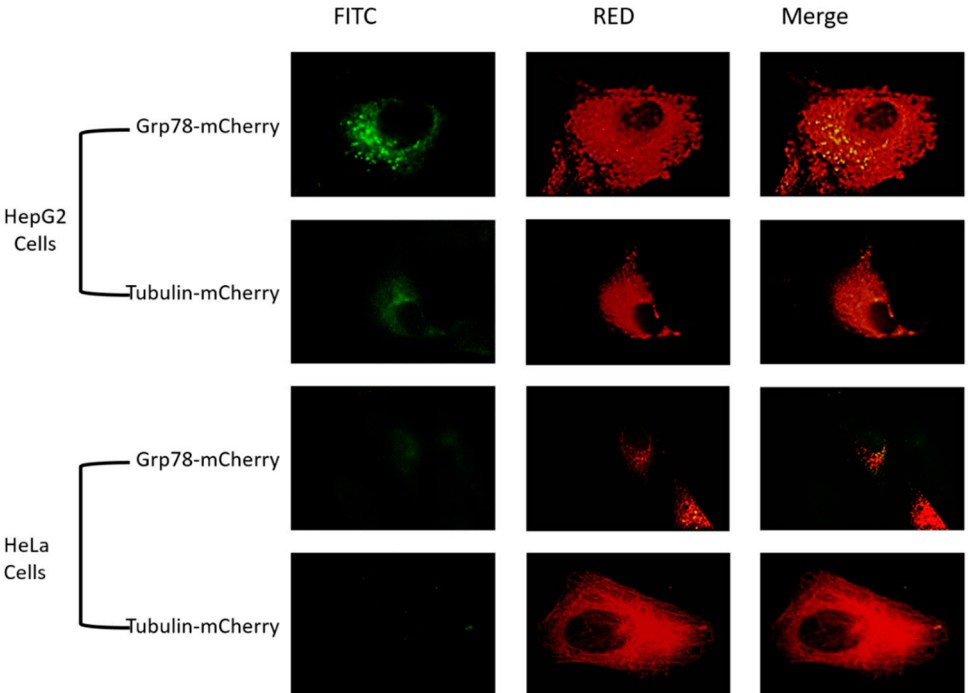

**Figure 5.** HepG2 and HeLa cells were transfected with Grp78-mCherry or tubulin-mCherry (Red) before they were incubated with SP94-core (fused with EGFP) protein at a concentration of 100 μM for two hours. The results revealed an increase in the efficacy of binding affinity on the cells that are transfected with plasmids expressing Grp78-mCherry compared to cells transfected with plasmids expressing tubulin-mCherry at magnification 100×.

## 4. Discussion

VLPs have been extensively utilized in vaccine development and therapeutic delivery, with promising preclinical and clinical results. In this study, HCV core protein fused with RGD and SP94 peptides were expressed by the baculovirus expression system to produce two fusion proteins; SP94-core protein and RGD-core protein. The optimum protein expression was obtained from infection Sf9 cells at MOI of 10 at 72 hpi. Although most of the proteins, especially transmembrane proteins, that are expressed by baculovirus are usually harvested at 48 hpi [16] before the compromization of the infected cells, we found 72 hpi to be more efficient in expressing the HCV core proteins based on the intensity of the fluorescence bands shown by in-gel fluorescence method.

The protein purification by NI-NTA affinity resin revealed that a small concentration of protein particles was obtained in elution, while a high concentration of protein was obtained in the flow-throw portion; this is perhaps due to the inaccessibility of the His-tag residues to NI-NTA. This inaccessibility may be due to the assembly nature of the core, one that may prompt interactions between the elements, fully or partially hiding

the His-tag. It is worth noting that the His-Tag was added to the C terminus of the HCV core, which is known to be involved in the proper folding [17]. Even with the addition of chaotropic compounds and/or detergents allowing total or partial solubilization, the presence of the C-terminal hydrophobic domain in the HCV core causes challenges with purification, specifically chromatographic support [18,19]. Yvon and his colleagues found difficulties in the purification of HCV core protein tagged with histidine in the C terminus, and opted to use electrophoresis techniques with a solubilization buffer compatible with electrophoresis. This buffer maintains an adequate amount of protein solubility during the separation process [20]. However, his-tag seems not to affect the oligomerization of the core, as the TEM results showed an average size of RGD-core VLPs and SP94-core VLPs in the range of 30–60 nm, which agreed with previously reported results of HCV-LP [6]. RGD peptides have some benefits for use in drug delivery applications; they can attach to different integrin species, which are highly expressed on cancer cell surfaces, especially $\alpha5\beta1$ and $\alpha v\beta3$ integrin [21]. In this study, we found that the intensity of the green fluorescence intensity, due to the binding affinity of RGD-core, was increased with HeLa cells and Huh7 cells more than HepG2 cells. That may be due to an increase in the expression of integrins levels especially ($\alpha5\beta1$ and $\alpha v\beta3$) in HeLa cells and Huh7 cells, compared to HepG2 cells, which are characterized by low expression of these two types of integrins [22–25]. In addition, we found that RGD-core protein was slightly bound with HEK-293T cells, which can be explained by the artificial immortalized nature of these cellsas they contain fragments of Adenovirus 5 DNA and a mutant version of SV40 large T antigen. Therefore, this slight binding may be due to the high presence of integrins on the HEK-293T cell surface [26]. A previous study revealed that flowcytometric analysis and immunocytochemistry results showed expression of different types of integrins on HEK-293T cells, such as $\beta3$, $\beta5$, $\alpha v\beta3$, $\alpha v\beta5$, $\alpha v\beta6$, $\beta1$, and $\alpha v$ [27]. SP94 is a peptide that targets HCC specifically, however, most studies used SP94 as a conjugated peptide to target liver cancer cells in vitro and in vivo [28–30]. The usage of SP94 as a conjugated chemical peptide is common, but it has some limitations. Synthetic peptides are now widely available and may be manufactured in huge quantities at a low cost. They are more resistant to enzymatic degradation and are more robust to pH and temperature fluctuations. However, synthetic peptides have some limitations which should be considered. Most of the chemically conjugated peptides are linked by polyethylene glycol (PEG), which has many side effects, such as obscuring the protein's surface, increasing the polypeptide's molecular size, lowering its renal traction, preventing antibodies or antigen-processing cells from recognizing the target cells, and slowing proteolytic enzyme destruction. In addition, PEG imparts the physicochemical properties of molecules, allowing them to be altered [31]. Therefore, our target in this study is to design genetically engineered peptides (SP94 and RGD) to avoid some limitations of chemically synthetic peptides on other nanocarriers. In this study, SP94 was designed as a genetically engineered peptide fused with EGFP-core, and it showed specificity for liver cancer cells HepG2 and Huh7 at a concentration of 100 µM, which is expected and in agreement with previous studies [28–30].

Grp78 is a multifunctional protein, which is mainly located in ER and activated after ER stress [32]. As mentioned above, Grp78 is a possible specific targeted receptor for SP94. Therefore, we wanted to confirm these previous results by increasing Grp78 expression in the different cell lines to investigate its role in the VLPs binding rate, as indicated by green fluorescence colocalization with Grp78, as shown in Figures 4 and 5. To determine whether Grp78 could play a role in enhancing the binding affinity of SP94-core and RGD-core proteins, Grp78-mCherry was overexpressed in HeLa cells and HepG2 cells; while overexpressing tubulin-mCherry was used as a control. We chose tubulin in this study as a control because it is an abundant cellular protein. A previous study revealed that there is a crosstalk between integrins and tubulin, which has an effect, and is attributed to RGD binding on the cell surface [33]. Our findings showed that overexpression of Grp78-mCherry in HepG2 cells increased the binding efficiency when incubated with SP94-core protein and RGD-core protein for 2 h compared to the overexpression of tubulin-mCherry,

which exhibited no change in the binding affinity of the two fusion proteins, as shown in Figures 4 and 5. In addition, we found that overexpression of Grp78-mCherry increased the binding affinity in HeLa cells after incubation with SP94-core protein and RGD-core protein, where HeLa cells are non-liver cancer cells, and SP94 peptide is specifically and exclusively targeting liver cancer cells, as shown in Figure 5. These results confirmed the sole role of GRP78 as an entry mediator for SP94 even in non-liver cancer cells. However, the RGD's increased binding efficacy may be due to the effect of overexpression of Grp78 on ECM, cell-cell adhesion, and increased expression of proteins in ECM, including integrins, which are the main target of the RGD peptides [34].

## 5. Conclusions

It is confirmed that the core-EGFP protein tagged with the SP94 peptide is specific for targeting liver cancer cells, while the VLP tagged with RGD peptide targets most types of cancer cells. The SP94 peptide works well as a fused peptide, not just as a conjugated peptide, as commonly used. In addition, it was confirmed that GRP78 is the main binding receptor for SP94 peptides, and can modulate integrins to increase RGD binding efficacy.

**Author Contributions:** Literature search, D.M., M.A.W. and T.Z.S.; Lab work and methodology design, D.M. and M.A.W.; data curation and analysis, D.M. and M.A.W.; original draft preparation, D.M.; writing review and editing D.M., M.A.W. and T.Z.S.; data acquisition, D.M., M.A.W. and T.Z.S., visualization D.M., M.A.W. and T.Z.S.; supervision, T.Z.S.; funding acquisition, T.Z.S. All authors have read and agreed to the published version of the manuscript.

**Funding:** This work is supported by The Egyptian Science, Technology & Innovation Funding Authority (STDF) (Grants Nos.: STDF 22976 and 45396).

**Institutional Review Board Statement:** Not applicable.

**Informed Consent Statement:** Not applicable.

**Data Availability Statement:** Not applicable.

**Acknowledgments:** We wish to thank the Center for Aging and Associated Diseases (CAAD) and the center of the genome (CG), Zewail City of Science and Technology, for allowing us to use their facilities. Moreover, we would like to thank all the lab members who participated in the initial work of the projects funded by STDF 22976 and STDF 45396.

**Conflicts of Interest:** Authors declare no conflict of interest.

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
