# Peer review of "Genetically Engineered Hepatitis C Virus-like Particles (HCV-LPs) Tagged with SP94 Peptide to Acquire Selectivity to Liver Cancer Cells via Grp78"

_cimb, doi:10.3390/cimb44080256_

Round 1

Reviewer 1 Report

Cancer targeting using virus like particles (VLPs) have been under investigation for quite some time and it has the potential to grow as a big industry in near future. In this context, the current article should draw considerable attention in the same and allied field of research. The authors in this paper claimed that their genetically engineered hepatitis C VLPs can specifically bind to liver cancer cells via Grp78. The background of the study was duly discussed with reasonable references, however, the data that were presented in support of the claim require more work to support the claims. Hence, I recommend the current article to be reconsidered after the following major revisions:

1.       For figures 2, 3, 4, 5, fluorescence intensity scale must be shown and all comparing images must be at same scale.

2.       In figure 2, the phase contrast and the FITC regions for the Hepg2 and Huh7 cells appears to be two different sections of the slide. The comparing phase contrast and FITC regions must be from the same region of the slides. Please also include a bright field-FITC overlay image for each cell lines in the panel. The FITC image for the HepG2 cell lines are smudged, indicating non-specific binding. Please revisit the experiment and if the images are still smudged, it cannot be called a specific binding by just this microscopic image. In such case, please include a more definitive evidence of specific binding, e.g flow cytometric analysis.

3.       Since no fluorescence scale was shown in figures 4 and 5, it is hard to rationally compare the different conditions describe in each of the figure panels. However, the red color needs to be tuned down a little so that the merges images can be more clean to visualize.

4.       In figure 2D, please use white color to mention the scale, the black writing are hard to see on the dark TEM background.

5.       In figure 2, both images D and E are out of focus, however, the image D can be used. The image E does not meet the minimum publication quality. Please provide a different image .

Thank you

Author Response

Comments from reviewer 1

Comment 1: For figures 2, 3, 4, 5, fluorescence intensity scale must be shown and all comparing images must be at same scale.

Response: Thanks a lot for your comment, we are not sure about the meaning of the florescence intensity scale, we will appreciate it if the review can elaborate more about this. If the reviewer mean to measure the whole intensity of each photo to compare it with the other photos, we think that this was not the aim of the study as we tried to detect whether the binding happened or not when we used the same parameters. We are planning in the future to optimize the protocol further but this needs to consider many parameters that are not the focus of this manuscript.

Comment 2:   In figure 2, the phase contrast and the FITC regions for the Hepg2 and Huh7 cells appears to be two different sections of the slide. The comparing phase contrast and FITC regions must be from the same region of the slides. Please also include a bright field-FITC overlay image for each cell lines in the panel. The FITC image for the HepG2 cell lines are smudged, indicating non-specific binding. Please revisit the experiment and if the images are still smudged, it cannot be called a specific binding by just this microscopic image. In such case, please include a more definitive evidence of specific binding, e.g flow cytometric analysis.

 Response:

The reviewer is right, therefore, we have added better images to indicate the specific binding as shown in figure 2 in the revised manuscript.  

We have done the overlay between phase contrast and FITC, and the results revealed that they are from the same regions as shown below.

Comment 3: Since no fluorescence scale was shown in figures 4 and 5, it is hard to rationally compare the different conditions describe in each of the figure panels. However, the red color needs to be tuned down a little so that the merges images can be cleaner to visualize.

Response: Thank you for your comment, the color red in figures 4 and 5 has already tuned down to visualize cleaner merged images as shown in the revised manuscript.

Comment 4: In figure 2D, please use white color to mention the scale, the black writing are hard to see on the dark TEM background.

Response: Thank you for your comment; I believe the reviewer meant Figure 1D. Both Figure 1D and 1E were replaced based on comment 5. Now the color of the scale bars are clear.

Comment 5: In figure 2, both images D and E are out of focus, however, the image D can be used. The image E does not meet the minimum publication quality. Please provide a different image.

Response: Thanks you for your comment, Agreed, therefore, both figures 2D and 2E have been replaced with better ones with clear scale as shown in the revised manuscript.

Reviewer 2 Report

This is an interesting article describing the possibility of targeting liver cancer cells with genetically modified HCV particles expressing specific peptides (SP94 and RGD). My major concern is that control experiments are missing with HCV particles expressing, for example, a nonsensical peptide motif (s.a., a scrambled version of SP94). In addition, the manuscript requires extensive improvements in language.

Figures 2 and 3: It is unclear to me what "Control" means. This should be explained in the legend.

Author Response

Comments from Reviewer 2

Comment 1: This is an interesting article describing the possibility of targeting liver cancer cells with genetically modified HCV particles expressing specific peptides (SP94 and RGD). My major concern is that control experiments are missing with HCV particles expressing, for example, a nonsensical peptide motif (s.a., a scrambled version of SP94). In addition, the manuscript requires extensive improvements in language.

Response: Thank you a lot for your valuable comments. In our study, the focus is the SP94 and we used the RGD as control to ensure the specificity of the SP94 to liver cells. We took in consideration the experiment design of a paper, published by Li et al (https://nanoscalereslett.springeropen.com/articles/10.1186/1556-276X-8-401#citeas), which did not use the core as a control. Therefore, since the specificity of the RGD was already confirmed with many studies, we used it as a control for the SP94 specificity.

In addition, the manuscript was rechecked intensively to improve its language as recommended. Changes are labelled in the revised manuscript in track changes.

Comment 2: Figures 2 and 3: It is unclear to me what "Control" means. This should be explained in the legend.

Response: We agree with your comment and explanation was added in fig. 2and 3 legends.

Reviewer 3 Report

Mofed, et al. investigated a novel drug delivery system based on HCV-LP using SP94 peptide for targeting liver cancer cells. They also examined an additional effect of Grp78 to enhance the binding capacity of SP94 to cancer cells.

Major

1.       Authors clearly showed that SP94 peptide targets liver cancer cells, not cancer cells originated from other organs. I am wondering whether SP94 peptide targets normal hepatocytes or not.

Minor

1.       FITC images in Figure 2 are not clear, especially HEK293T and HepG2.

2.       Reference #28 should be revised as “Scientific Reports. 2016,6,33511.”.

3.       Authors should check if Reference #29 is correct to use here because #29 did not say anything about SP94.

Author Response

Comments from Reviewer 3

      Comment 1:  Authors clearly showed that SP94 peptide targets liver cancer cells, not cancer cells originated from other organs. I am wondering whether SP94 peptide targets normal hepatocytes or not.

Response: Thanks a lot for your comment. In our study, we focused on the role of SP94 as a specific target peptide for liver cancer cells. Many previous studies, focused on the role of SP94 only on liver cancer cells such as

https://pubs.acs.org/doi/10.1021/acsabm.0c01468

 https://www.ncbi.nlm.nih.gov/pmc/articles/PMC7997290/

https://www.nature.com/articles/srep33511

In addition, we tried several times to obtain normal hepatocyte cells but we could not get them in our region. However, we use HEK293T cells instead, despite the fact that they are engineered and have some common characteristic with cancer but they are not cancer cells and we used them as a reference.

Comment 2:   FITC images in Figure 2 are not clear, especially HEK293T and HepG2.

Response: Thank you for your comment. We tried to use images with better resolution for HEK293T and replaced HepG2 photos with better ones.

Comment 3:  Reference #28 should be revised as “Scientific Reports. 2016,6,33511.”

Response: Thanks for your comment. Agree. We have revised it as Scientific Reports. 2016,6,33511, as requested.

Comment 4: Authors should check if Reference #29 is correct to use here because #29 did not say anything about SP94.

 Response: Thanks a lot for your valuable notice. This was a mistake in the references section and we corrected it. Therefore, we changed the reference of #29  with  “Chopra, M., Sgro, A., Norret, M., Blancafort, P., Iyer, K., Evans, C. SP94-Targeted Nanoparticles Enhance the Efficacy of Sorafenib and Improve Liver Cancer Cell Discrimination. ACS Applied Bio Materials. 2020, 4, 1023-1029”.

Instead of “Eso, Y.; Marusawa, H.Novel approaches for molecular targeted therapy against hepatocellular carcinoma. Hepatology Research. 2018, 488, 597-607. Page no. 12, line no 554.

Round 2

Reviewer 2 Report

Thank you for responding to my comments. The manuscript can now be published. 

Reviewer 3 Report

The authors adequately answered my question.